# Neutrophil Extracellular Traps Promote Metastases of Colorectal Cancers through Activation of ERK Signaling by Releasing Neutrophil Elastase

**DOI:** 10.3390/ijms24021118

**Published:** 2023-01-06

**Authors:** Michio Okamoto, Rei Mizuno, Kenji Kawada, Yoshiro Itatani, Yoshiyuki Kiyasu, Keita Hanada, Wataru Hirata, Yasuyo Nishikawa, Hideyuki Masui, Naoko Sugimoto, Takuya Tamura, Susumu Inamoto, Yoshiharu Sakai, Kazutaka Obama

**Affiliations:** 1Department of Surgery, Graduate School of Medicine, Kyoto University, Kyoto 606-8501, Japan; 2Department of Surgery, Uji-Tokushukai Medical Center, Kyoto 611-0041, Japan; 3Department of Surgery, Kurashiki Central Hospital, Okayama 710-8602, Japan; 4Rogel Cancer Center, University of Michigan, Ann Arbor, MI 48109, USA; 5Department of Surgery, Japanese Red Cross Osaka Hospital, Osaka 543-8555, Japan

**Keywords:** neutrophil extracellular traps, colorectal cancer, neutrophil elastase, ERK

## Abstract

Neutrophil extracellular traps (NETs) play important roles in host immunity, as there is increasing evidence of their contribution to the progression of several types of cancers even though their role in colorectal cancers (CRCs) remains unclear. To investigate the clinical relevance of NETs in CRCs, we examined the expression of citrullinated histone H3 using immunohistochemistry and preoperative serum myeloperoxidase–DNA complexes in CRC patients using an enzyme-linked immunosorbent assay. High expression of intratumoral or systemic NETs was found to correlate with poor relapse-free survival (RFS), for which it is an independent prognostic factor. In vitro investigations of CRC cells (HCT116, HT29) revealed that NETs did not affect their proliferation but did promote the migration of CRC cells mediated by neutrophil elastase (NE) released during NETosis to increase extracellular signal-regulated kinase (ERK) activity. In vivo experiments using nude mice (KSN/slc) revealed that NE inhibition suppressed liver metastases in CRC cells, although it did not affect the growth of subcutaneously implanted tumors. Taken together, these results suggest that NET formation correlates with poor prognoses of patients with CRC and that the inhibition of NE could be a potential therapy for CRC metastases.

## 1. Introduction

Colorectal cancer (CRC) is the third most common malignant disease following breast and lung cancer worldwide [1]; fatalities have increased by more than 30% over the past 15 years and are expected to increase by 25% over the next 10 years despite advances in surgical techniques, radiotherapy, chemotherapy and molecular-targeted drugs [2]. At least one-third of CRC patients develop liver metastases, and CRC-related death is usually attributable to distant metastasis [3,4]. Once the disease spreads to distant organs, neither conventional chemotherapy nor current targeted therapy offers significant benefits. Therefore, it is important to understand the mechanisms that promote cancer progression to overcome the distant metastases of CRCs.

It has been reported that several types of host cells, including macrophages, fibroblasts and mesenchymal stem cells, play important roles in the formation of the tumor microenvironment (TME) to support cancer progression [5,6,7]. The cross-talk between cancer cells and the components of TME mediated by TGF-β, TNF-α, TNF-β and NF-kB signaling contribute to cancer progression [8,9]. Additionally, in the context of TME, recent studies have shown that some populations of neutrophils, known as tumor-associated neutrophils (TANs), promote the growth, invasion, angiogenesis and metastasis of cancer cells, although they have been classically considered to exhibit defensive effects against tumor cells [10]. In fact, an increase in the neutrophil-to-lymphocyte ratio (NLR) in the peripheral blood has been reported to correlate with poor clinical outcomes in various types of cancers, including pancreatic, gastric, breast and colorectal cancers [11,12,13,14,15,16]. TANs recruited to the tumor site, mediated by C-X-C motif chemokine receptor 2 (CXCR2) and its ligands (i.e., CXCL1, CXCL2, CXCL3, CXCL5, CXCL7 and CXCL8), contribute to tumor progression by secreting matrix metalloproteinase-9 (MMP-9), vascular endothelial growth factor (VEGF), arginase, hepatocyte growth factor and numerous chemokines, such as CCL2, CCL5 and CXCL4, which can exert paracrine effects on the TME [17,18].

Recent studies have highlighted the involvement of neutrophil extracellular traps (NETs) in promoting cancer cell progression. These are extracellular web-like structures of DNA associated with granular proteins, including myeloperoxidase (MPO), neutrophil elastase (NE) and MMPs, released from activated neutrophils to trap and neutralize pathogens during the innate immune response [19]. Among them, NE, a serine protease, which has a high affinity for DNA, is suggested to play an important role in NET formation [20]. In the TME, NETs have been suggested to physically capture circulating tumor cells, resulting in the promotion of the migration of cancer cells to metastatic sites [21,22]. Furthermore, granular proteins released from neutrophils activate signaling pathways, including mitogen-activated protein kinase (MAPK) signaling, NF-κB, focal adhesion kinase (FAK)/extracellular signal-regulated kinase (ERK) signaling and ILK-β-parvin pathways, which promote cancer progression [23]. Notably, ERK, which is involved in the Ras-Raf-MEK-ERK signal transduction cascade, plays an important role in the regulation of various biological events, including cell cycle progression, cell adhesion, migration, proliferation, differentiation and transcription [24]. Recent studies have revealed the contribution of NETs to the progression of lung, esophageal and pancreatic cancers [25,26,27,28,29]; however, the relationship between NETs and CRCs is not fully understood.

Therefore, in this study, we aimed to investigate the clinical relevance of NETs in CRCs and the effect of NETs on CRC cells and its underlying mechanisms, and to explore the possibility of NETs as a potential therapeutic target for CRCs. We found that the high expression of intratumoral and systemic NETs in CRC patients without distant metastases correlates with poor relapse-free survival (RFS), of which it is an independent prognostic factor. We also revealed that NE released during NET formation accelerated the migration of CRC cells through the activation of ERK in vitro. Additionally, we demonstrated that the inhibition of NE in a liver metastatic mouse model significantly decreased the formation of liver metastases of CRC cells, suggesting that the inhibition of NE could be a potential therapeutic target for CRC metastasis.

## 2. Results

### 2.1. NET Formation Correlates with Poor Prognoses in CRC Patients

To evaluate the clinical relevance of NETs in CRCs, we examined the expression of neutrophil markers (i.e., MPO, MMP-9 and NE) and a marker of NETs, citrullinated histone H3 (Cit-H3), via the immunohistochemical (IHC) analysis of 133 patients with stage I, II and III CRC who underwent curative resection. Neutrophils infiltrating CRC tissue were detected by staining for MPO, MMP-9 and NE. The expression of Cit-H3 colocalized with the expression of these neutrophil markers, indicating the induction of NETs in CRC tissues (Figure 1A). The patients were divided into two groups according to the expression level of Cit-H3: the high Cit-H3 group (*n* = 67) and the low Cit-H3 group (*n* = 66). No significant differences in patient characteristics were observed between the two groups (Appendix A). We then analyzed overall survival (OS) and RFS and found that the high expression of Cit-H3 was significantly correlated with poor RFS (*p* = 0.009). In addition, there was a tendency for high Cit-H3 expression to correlate with poor OS, although this was not statistically significant (*p* = 0.054) (Figure 1B).

Next, we investigated potential prognostic factors for RFS (Table 1). Univariate analysis showed that poor RFS was significantly associated with the *n* factor, lymphatic invasion, preoperative therapy and high expression of Cit-H3. For multivariate analysis using the Cox proportional hazard regression model, the *n* factor (hazard ratio [HR] 3.26, 95% confidence interval [CI] 1.38–7.70, *p* = 0.0071) and high expression of Cit-H3 (HR 2.86, 95% CI 1.25–6.54, *p* = 0.013) remained significantly associated with poor RFS. We also investigated potential prognostic factors for OS (Appendix A). Univariate and multivariate analyses using the Cox proportional hazard regression model indicated that preoperative therapy (HR 3.94, 95% CI 1.45–10.88, *p* = 0.0073) was significantly associated with poor OS, and that high expression of Cit-H3 tended to be associated with poor OS (*p* = 0.12).

To further evaluate the clinical significance of NETs in CRC patients, we measured the preoperative serum level of MPO–DNA in 67 patients with stages II and III CRC, as serum MPO–DNA is a reliable marker of systemic NET formation [27]. The patients were assigned to two groups based on the average level of serum MPO–DNA in eight healthy volunteers: a high MPO–DNA group (*n* = 31) and a low MPO–DNA group (*n* = 36). There was no significant difference in patient characteristics between the groups, except for a higher percentage of patients with colon cancer in the high MPO–DNA group (Appendix A). Next, we compared the OS and RFS between the groups and found that a high preoperative serum level of MPO–DNA was significantly associated with poor RFS (*p* = 0.018). No significant difference was observed in OS (*p* = 0.43) (Figure 1C).

We then investigated potential prognostic factors for RFS in this cohort (Table 2). Univariate and multivariate analyses showed that sex (HR 0.26, 95% CI 0.10–0.72, *p* = 0.009), venous invasion (HR 5.69, 95% CI 1.58–20.51, *p* = 0.008), high CA19-9, a tumor marker, levels (HR 3.16, 95% CI 1.07–9.22, *p* = 0.036) and high MPO–DNA levels (HR 3.53, 95% CI 1.31–9.56, *p* = 0.013) were significantly associated with poor RFS. We also investigated potential prognostic factors for OS (Appendix A). For univariate and multivariate analyses, location (0.11, 95% CI 0.014–0.89, *p* = 0.038) and higher CA19-9 levels (HR 18.63, 95% CI 1.13–71.27, *p <* 0.0001) were significantly associated with poor OS. Taken together, these results indicate that intratumoral and systemic NET formation correlates with poor prognoses in patients with CRC.

### 2.2. NETs Accelerate the Migration of CRC Cells through the Release of NE

To elucidate the role of NETs in the progression of CRCs, we evaluated their effects on human CRC cell lines in vitro. For this purpose, we used human neutrophils isolated from healthy volunteers and the human CRC cell lines HCT116 and HT29. To induce NETs in vitro, human neutrophils were treated with phorbol-12-myristate-13-acetate (PMA), a reagent known to induce NET formation [30]. Isolated human neutrophils were incubated with SYTOX green to detect NETs by staining for extracellular DNA. We found that PMA treatment successfully induced NET formation in approximately 25% of neutrophils, whereas almost no neutrophils were stained with SYTOX green in the control, which is consistent with previous reports [19,31,32] (Appendix A). NETs have been reported to promote cancer progression by releasing extracellular DNA or neutrophil granular proteins [23,29]. To investigate the effects of the substances released during NET formation, we collected the supernatant of PMA-treated human neutrophils, which was used as an NET-conditioned medium (NET-CM) in the following experiments.

First, we assessed the effect of NET-CM on CRC cell proliferation; however, no significant effects were observed (Figure 2A). We also analyzed its effect on the migration of CRC cells using the wound healing assay and found that NET-CM significantly accelerated the migration of CRC cells compared to that of the control medium (Figure 2B and Appendix A). The effect of NE, which is essential for NET formation [33], on the migration of HCT116 and HT29 cells was evaluated, and NE was found to significantly accelerate migration (Figure 2C). This acceleration was inhibited by the administration of the NE inhibitor sivelestat. Interestingly, the enhanced migratory ability of CRC cells induced by NET-CM was also inhibited by the administration of sivelestat (Figure 2C). These results suggest that NE plays an essential role in the mechanism by which NETs accelerate CRC cell migration.

### 2.3. ERK Is an Important Regulator of Activated Cell Migration Induced by NETs

Next, we investigated the molecular mechanisms by which NE activates the migration of CRC cells. Previous studies have reported that ERK is a key regulator of cell migration in various types of cells [34,35]; therefore, we investigated whether accelerated CRC cell migration caused by NET-CM or NE is regulated by ERK. To visualize the ERK activity using time-lapse live imaging [36], we transduced a Förster resonance energy transfer (FRET, Île-de-France, France) biosensor for ERK (pPBbsr2-3560 NES) into HCT116 cells. ERK activity is presented by FRET/CFP ratio images (Appendix A).

ERK activity in HCT116 cells was dramatically elevated within 20 min after the administration of NET-CM or NE, which was blocked by sivelestat treatment (Figure 3A–C). The accelerated migratory ability of HCT116 and HT29 cells induced by NET-CM or NE treatment in the wound healing assay was blocked by the administration of a mitogen-activated protein kinase kinase (MEK) inhibitor (PD0325901), which suppresses ERK activity (Figure 3D). These results suggest that ERK plays an important role as a downstream molecule of NET-CM or NE in the enhanced migration of CRC cells.

### 2.4. The Inhibition of NE Suppresses NET Formation Resulting in Decreased Liver Metastases of CRC Cells

As in vitro experiments indicated that NE plays an important role in the enhanced migration of CRC cells via NETs through the activation of ERK, we evaluated the therapeutic potential of NE inhibition using in vivo experiments.

First, to assess the effect of NE on the proliferation of CRC cells in vivo, we transplanted HCT116 cells into the dorsal flanks of 7–8-week-old female KSN/slc nude mice and evaluated the effect of sivelestat treatment on tumor growth. Mice in the control and sivelestat groups were subcutaneously injected with HCT116 cells that were pre-incubated with NET-CM or NET-CM plus sivelestat, respectively. Mice were administered with a daily intraperitoneal injection of the vehicle and sivelestat (10 mg/kg, CAYMAN) from the day before the transplantation of HCT116 cells. No significant change in tumor size was detected between the two groups after five weeks treatment, although the successful suppression of NETs in the sivelestat group was confirmed by the decreased expression of Cit-H3 (Figure 4A–D). As expected, no significant difference was observed in the proportion of Ki67-positive cells in xenografts (Figure 4E). These results support the in vitro finding that NETs have no effect on the proliferation of CRC cells.

Next, we evaluated the effect of NE on CRC cell metastasis using the experimental liver metastasis model. Luciferase-expressing HCT116 cells (HCT116-Luc) were injected into the spleens of nude mice, which enabled the monitoring and quantification of metastasized tumor cells in the liver with bioluminescence [37,38]. Nude mice were administered with a daily intraperitoneal injection of 10 mg/kg sivelestat or the vehicle from the day before the splenic injection of HCT116-Luc cells. The bioluminescence intensity started to increase soon after the injection of HCT116-Luc cells in the control group, whereas it started to increase two weeks later in the sivelestat group. A significant decrease in bioluminescence intensity was observed in the sivelestat group compared to the control group after two weeks of injection (*p <* 0.05) (Figure 5A,B). Interestingly, the bioluminescence signal in both groups increased at similar rates after these two weeks (Figure 5A). Mice were sacrificed five weeks later, and the decrease in liver metastases was confirmed macroscopically in the sivelestat group (Figure 5C). IHC analysis revealed the decreased expression of Cit-H3 in liver metastases of sivelestat-treated mice, indicating the successful suppression of NET formation. However, no significant change in the proportion of Ki-67-positive cells was detected between the two groups (Figure 5D–F). These results indicate that the inhibition of NE suppresses the infiltration of tumor cells into liver tissues from the venules, which is the initial step of liver metastasis. This could explain the delay in the increase in bioluminescence in the first two weeks after CRC cell injection. However, once tumor cells extravasate into the liver tissues, NE inhibition does not suppress the proliferation of CRC cells, which was shown by a similar increase in bioluminescence after two weeks of injection. These results indicate that the inhibition of NE by sivelestat could be a potential therapeutic option to suppress the liver metastasis of CRC cells (Figure 5G).

## 3. Discussion

NETs are extracellular web-like structures built from nuclear or mitochondrial DNA fibers complexed with histones and granular proteins [19]. During NET formation, neutrophils release eight types of proteins, including highly homologous serine proteases, such as NE, cathepsin G, azurocidin and MPO [39]. Although NETs were originally considered to be released for entrapping pathogens as one of the mechanisms to protect organisms from foreign harmful microbes [40], its additional functions in promoting tumor progression as TANs have been revealed recently. In response to external stimuli (e.g., pathogenic microorganisms and their derivatives, physicochemical stimulation, inflammatory cytokines and metabolites), neutrophils collapse and release DNA in the form of webs, which physically trap tumor cells and contribute to their retention in the capillaries of metastases [23]. NETs can also wrap and coat cancer cells, shielding them from clearance by immune cells [41]. The released cytokines and granular proteins, such as NE and MMPs, can promote the progression of cancer cells through the degradation of the extracellular matrix (ECM) and the induction of pro-tumorigenic signaling pathways [42]. The proteases released during NETosis can induce the remodeling of laminin, which awakens dormant tumor cells and triggers the integrin signaling pathway [43]. It was reported that NETs contribute to the formation of arterial, venous and cancer-associated thrombosis, which protect cancer cells from shear forces and assault by immune cells [44].

The relationship between NETs and patient prognosis has been reported in several types of cancer. Zhang et al. evaluated the infiltration of NETs in esophageal cancer using IHC in 126 patients who underwent esophagectomy and demonstrated that a higher level of NET infiltration was associated with poor OS and disease-free survival (DFS) [45]. Rayes et al. measured the expression levels of circulating MPO–DNA in patients with esophageal and lung cancers and revealed that the expression levels of circulating MPO–DNA correlated with the disease stage [46]. Zhang et al. evaluated NETs in the peripheral blood of patients with gastric cancer and demonstrated that they had diagnostic, therapeutic, predictive and prognostic values [47]. Although our study has several limitations, patients were recruited from a single center, and the clinical analysis was a non-randomized retrospective study. We found that patients with high expression of intratumoral or systemic NET formation exhibited significantly poor RFS and that they were independent predictive factors for poor prognosis by multivariate analyses. The patient cohort in this study excluded patients with stage IV CRC. In contrast, Yazdani et al. focused on stage IV CRC patients with liver metastases and reported that the expression levels of preoperative MPO–DNA in CRC patients were significantly higher than those in healthy controls. They also showed that the OS and DFS of patients with high MPO–DNA expression were poorer than those of patients with low expression [48]. Taken together, NETs can contribute to the progression of CRCs as well as to other types of cancer.

We demonstrated that the inhibition of NE suppresses the liver metastasis of CRC cells using an experimental animal model. Although NE was originally considered to have a function in clearing pathogens during infection [49], the tumorigenic functions of NE have been revealed in lung [50], colon [51] and breast cancers [52,53,54]. As NE is an integral component of NETs and is also required for NETosis, NE inhibition can suppress NET formation. In fact, sivelestat treatment successfully suppressed NET formation in both subcutaneously implanted tumors and liver metastases in the present study; however, it did not inhibit the growth of subcutaneously implanted tumors, whereas it significantly suppressed liver metastases. These results suggest that the inhibition of NE suppressed the initial step of the liver metastasis of CRC cells. The fact that sivelestat treatment could not suppress the growth of liver metastases once the CRC cells metastasized to the liver (Figure 5A) supports this notion. Considering the in vitro results showing that NET-CM promoted migration but did not affect the proliferation of CRC cells (Figure 2), one possible explanation of decreased liver metastases by sivelestat treatment could be attributed to the suppression of CRC cell migration. It was reported that cancer cells are exposed to attacks from immune cells such as natural killer (NK) cells at the metastatic sites [55]. Another possible mechanism of decreased liver metastases is that NE inhibition could allow CRC cells to escape from elimination by immune cells. Further investigation is needed to clarify the underlying mechanism.

In the present study, we revealed that NE activates ERK in CRC cells, which plays important roles in various biological events, including cell proliferation, differentiation and migration [56]. NE is reported to cleave pro-transforming growth factor-α (pro-TGF-α) from the cell membrane, resulting in the phosphorylation of the epidermal growth factor receptor (EGFR), which triggers the ERK signaling pathway [57]. Previous studies have also shown that the migration of cancer cells is accelerated by NETs through the induction of epithelial-to-mesenchymal transition (EMT) in cancer cells [25,58], the remodeling of the stroma to facilitate cell spread and the activation of the protease cascade [59] and ILK-β parvin signaling pathway [60].

However, the effects of NETs on cell proliferation remain controversial. In the present study, we showed that NETs did not affect the proliferation of CRC cells in vitro and in vivo despite the fact that NETs upregulated ERK activity in vitro in CRC cells. Additional factors involved in NETosis may also be involved in the regulation of cell proliferation. Our results are similar to some previous studies that have shown that NETs do not accelerate cell proliferation but promote cell migration in ovarian, pancreatic, head and neck cancers [28,61,62]. However, some studies have demonstrated that NETs promote the proliferation of cancer cells [63]. The involvement of NF-κB signaling and FAK/ERK/MLCK/YAP signaling was previously reported [43]. Yazdani et al. reported that NE inhibition significantly suppresses the growth of subcutaneously implanted tumors in CRC cells [48]. They used an NE inhibitor, GW31161, whose activity is stronger than that of sivelestat, which could explain the discrepancy between their results and ours [64]. Nawa et al. showed that the effect of sivelestat on cell proliferation is dose-dependent [65]. Therefore, it is possible that an adequate concentration of sivelestat suppressing cell proliferation may not be obtained in our experimental settings. Further investigation of the effects of NETs on cell proliferation is required.

Although clinical trials have not yet been conducted, previous studies using experimental animal models have suggested the effectiveness of NET-targeting therapies for several types of cancers. The therapeutic potential of DNase I has been demonstrated in the liver metastases of CRC [27,48], lung cancer [22,26,46] and pancreatic cancer [28]. Park et al. demonstrated that the inhibition of peptidyl arginine deiminase 4 (PAD4) suppresses the migration and invasion of breast cancer cells, resulting in decreased lung metastases [66]. Lee et al. reported that NETs contribute to the formation of the premetastatic omental niche of ovarian cancers and that the blockade of NET formation by PAD4 inhibition significantly decreases omental metastases [62]. Xiao et al. demonstrated that the tumor-secreted protease cathepsin C (CTSC) promotes the recruitment of neutrophils and NET formation, resulting in the increased lung metastasis of mouse breast cancer cells. They showed that the inhibition of CTSC suppresses NET formation, resulting in decreased lung metastases without affecting the growth of primary tumors [67]. We demonstrated that sivelestat treatment significantly suppresses NETosis and decreases liver metastases. This result supports the notion that NE inhibition could be a potential therapeutic strategy to suppress the liver metastases of CRCs. Based on the finding that sivelestat treatment suppresses the initial process of metastasis, it is important to combine sivelestat with conventional chemotherapy, which can suppress the proliferation of CRC cells. As sivelestat has already been approved for the treatment of acute respiratory distress syndrome [68], it could be a potential therapeutic option to suppress the metastasis of CRCs. Clinical trials of sivelestat in patients with CRC are expected.

This study has several limitations. First, in the clinical analysis, patients were recruited from a single center, and the sample size was relatively small. Second, we focused on the effect of NE among the granular proteins released during the process of NET formation. Additional analyses of the effects of other granular proteins on CRC would be necessary to reveal the detailed mechanisms by which NETs promote CRC progression.

In conclusion, we demonstrated that NET formation correlates with poor prognoses in patients with CRC and is an independent predictor of poor prognoses in them using multivariate analysis. NE released during NET formation accelerates the migration of CRC cells through the activation of ERK. Additionally, we demonstrated that the inhibition of NE in a liver metastatic mouse model significantly decreases the formation of the liver metastases of CRC cells, suggesting that NE inhibition could be a potential therapeutic strategy for CRC metastasis. Clinical trials of NET-targeting therapy for CRCs are expected.

## 4. Materials and Methods

### 4.1. Patient Population

A total of 133 patients with pathological stage I–III CRC underwent primary resection at Kyoto University Hospital between January 2006 and December 2007, and their tissue samples were retrospectively analyzed using IHC. For the analysis of serum MPO–DNA levels using the enzyme-linked immunosorbent assay (ELISA), preoperative serum samples were collected from 67 pathological stage II–III CRC patients at Kyoto University Hospital between November 2011, and February 2014, and eight healthy volunteers served as controls. The study protocols were approved by the institutional review board of Kyoto University (approval number; R2908-2), and the patients provided their consent for data analysis.

### 4.2. Immunohistochemical Analysis

Tissue blocks of formalin-fixed paraffin-embedded surgical specimens of CRCs were sectioned into 4 μm slices for IHC. Following antigen retrieval, tissue sections were incubated with respective primary antibodies (Appendix A) overnight at 4 °C and were stained using the avidin-biotin immunoperoxidase method. The presence of the tumor was confirmed by H&E staining. For the detection of infiltrating NETs, intratumoral and peritumoral tissues were observed, and the number of Cit-H3-positive cells per high-power field was independently counted by two researchers in a blinded manner. The average numbers of Cit-H3-positive cells in the four fields of view was recorded as the number of Cit-H3-positive cells per sample. Slides with different evaluations among the two researchers were reinterpreted at a conference to reach a consensus. Based on the number of Cit-H3-positive cells, CRC patients were classified into two groups: high Cit-H3 and low Cit-H3 groups, according to the median value. Representative images were obtained using a fluorescence microscope (BZ-X810; Keyence, Osaka, Japan).

### 4.3. ELISA

MPO–DNA complexes have been identified using a capture ELISA [60,69,70]. As the capturing antibody, 75 µL of anti-MPO monoclonal antibody (5 µg/mL, ABD Serotec, Oxford, UK) was applied to 96-well microtiter plates and was incubated overnight at 4 °C, sealed with a film cover. After blocking with 1% bovine serum albumin (BSA, Irving, TX, USA), 40 µL of patient serum was added per well in combination with peroxidase-labeled anti-DNA monoclonal antibody (component No. 2 of cell death detection ELISA kit; Roche, Basel, Switzerland) according to the manufacturer’s protocol. After 2 h of incubation at room temperature (RT) on a shaking device (320 rpm), the samples were washed three times with 200 µL of phosphate-buffered saline (PBS, Boston, MA, USA) per well, and the peroxidase substrate (ABTS of cell death detection ELISA kit; Roche) was added. The absorbance at a wavelength of 405 nm was measured using a GloMax Navigator System with DI and PS (Promega, Madison, WI, USA) after 40 min of incubation at 37 °C under light shielding.

### 4.4. Cell Lines and the Isolation of Human Neutrophils

CRC cell lines, HCT116 and HT29, were obtained from the American Type Culture Collection (Manassas, VA, USA). These cell lines were maintained in low-glucose DMEM supplemented with 10% fetal bovine serum (FBS) and 1% penicillin/streptomycin. HCT116 cells stably expressing firefly luciferase (JCRB1408) were obtained from the Japanese Collection of Research Bioresources Cell Bank (Osaka, Japan) and were subjected to single-cell cloning [37]. Human neutrophils were isolated from healthy donors using the MACSxpress Whole Blood Neutrophil Isolation Kit, human (Miltenyi Biotec, Bergisch Gladbach, Germany), according to the manufacturer’s protocol [71].

### 4.5. In Vitro NET Formation Assay

Neutrophils isolated from healthy donors were plated in the Cell Cultivation Flask (VTC-F75V; AS ONE) with RPMI1640 (Nacalai tesque, Kyoto, Japan) containing 1% BSA for 1 h before stimulation with 100 nM Phorbol-12-myristate-13-acetate (PMA; Sigma-Aldrich, St. Louis, MO, USA). After 2 h of PMA stimulation at 37 °C and 5% CO_2_, the cell culture medium was centrifuged at 300× *g* for 5 min and was resuspended in a phenol red-free and serum-free medium containing HEPES (DMEM/Ham’s F12; Nacalai tesque). After 6 h of incubation, the medium was centrifuged at 300× *g* for 5 min, and the supernatant was collected and used as an NET-conditioned medium (NET-CM) for subsequent experiments. The neutrophil control medium was composed of the same medium in which neutrophils were cultured for 6 h without PMA stimulation.

### 4.6. Cell Proliferation Assay

Cell proliferation was assayed with Cell Counting Kit-8 (Dojin, Kumamoto, Japan) according to the manufacturer’s protocol [72,73]. Cells were plated at a density of 2000 cells/well in 96-well plates in serum-free media (DMEM; Nacalai tesque) with the vehicle, neutrophil control medium or NET-CM. The absorption spectrum of each sample was analyzed using GloMax–Multi + Detection System (Promega) on days 0, 1, 2 and 3, and the relative rate with respect to the day 0 value was calculated as 1.

### 4.7. Cell Migration Assay Using Scratch Wound Healing

Six well plates were coated with 0.3 mg/mL Cellmatrix Type I-C (Nitta Zelatin, Yaoshi, Japan) and were incubated for 2 h at 37 °C. After incubation, Cellmatrix was removed, and the coated plates were blocked with 0.3% BSA for an hour at 37 °C. After washing the plates with PBS, 5 × 10^5^ HCT116 and HT29 cells per well were plated and incubated for 24 h. The plates were scratched with a 1000 μL pipette tip and were washed with PBS. FBS-free medium (DMEM Ham’s F12; Nacalai Tesque), neutrophil control medium and NET-CM were added to each well. Furthermore, cells were treated with NET-CM, NET-CM + 100 µM sivelestat (17779; Cayman, Ann Arbor, MI, USA), NET-CM + 10 nM MEK inhibitor (PD0325901; Chemscene, Monmouth Junction, NJ, USA), 10 µg/mL NE (ENZO, Farmingdale, NY, USA), 10 µg/mL NE + 100 µM sivelestat and the vehicle control. The doses of reagents were determined according to previous studies [74,75,76]. Wounds were observed with a phase-contrast microscope (BZ-X810; Keyence) to measure the widths of six points in each wound. The averaged value of these six points was used as the width of each wound.

### 4.8. FRET Imaging of ERK Activity

The FRET biosensor for ERK (pPBbsr2-3560 NES) was reported previously [77]. To establish stable cells expressing FRET biosensors, we used the PiggyBac transposon-mediated gene transfer. The pPBbsr2-3560 NES was transfected with pCMV-mPBase into HCT116 cells using Lipofectamine 2000. After blasticidin S (Nacalai Tesque) selection at 10 µg/mL for a week, the bright cell population was collected using fluorescence-activated cell sorting (FACSAria II; BD Biosciences, Franklin Lakes, NJ, USA). An amount of 2 × 10^5^ cells of HCT116-pPBbsr2-3560NES were plated on 35 mm glass-bottom dishes (D11141H; MATSUNAMI) coated with collagen type I. After 24 h, the cells were serum-starved for 6 h in DMEM/Ham’s F12 supplemented with 0.1% BSA (Sigma-Aldrich). Cells were treated with NET-CM, NET-CM + 100 µM sivelestat, 10 µg/mL NE, 10 µg/ml NE + 100 µM sivelestat or the vehicle control. Confocal fluorescence images were acquired before and 20 min after the drug treatment using an IX83 inverted microscope (Olympus, Tokyo, Japan) equipped with an UPlanSApo 20X or 40X objective lens (Olympus), DOC-Cam 2736 × 2192 (4.54 × 4.54 µm), DOC-Cam HR-M (Molecular Devices, San Jose, CA, USA), lumencor Spectra X light engine (Lumencor, Beaverton, OR, USA), an IX2-ZDC laser based autofocusing system (Olympus) and a BIOS-425T-OL automatically programmable XY stage (SIGMA KOKI, Tokyo, Japan). The stage-top incubator INUG2F-IX3W was used to maintain the temperature, humidity and CO_2_ concentration. The following filters were used for dual-emission imaging: an exciter (CFP, Washington, DC, USA), 438/24 SPECTRA7/Exiter, dichroic mirror (FRET) FF458-Di02-25 × 36 (Semrock, West Henrietta, NY, USA), emitter (CFP) FF01-483/32-25 (Semrock) and emitter (YFP) FF01-542/27-25 (Semrock). Acquired images were analyzed using the MetaMorph software (Universal Imaging, Miami Lakes, FL, USA), as described previously [78,79]. In brief, the FRET level was represented by the FRET/CFP ratio image in the intensity-modulated display mode. Eight colors from red to blue were used to represent the FRET/CFP ratio, and 32 grades of color intensity were used to represent the signal intensity of the CFP image. Warm and cold colors indicate high and low FRET levels, respectively.

### 4.9. In Vivo Subcutaneous Implanted Tumor Models

HCT116 cells incubated in NET-CM overnight with or without the NE inhibitor sivelestat were used in both the groups, respectively. A total of 2 × 10^6^ stimulated HCT116 cells were suspended in 50 µL PBS in addition to 50 µL Matrigel (Corning, Somerville, MA, USA) and were subcutaneously injected into the dorsal flanks of 7–8-week-old female KSN/slc nude mice (Japan SLC, Shizuoka, Japan) in each group. Mice in the sivelestat and control groups were injected intraperitoneally with 10 mg/kg of sivelestat and the vehicle every 24 h from the day before cell inoculation until they were sacrificed, respectively. The dose of sivelestat was determined according to previous studies [74,75,76,80]. Treatment started the day before inoculation. Tumor sizes were measured with calipers once a week, and the tumor volumes were estimated using the following formula: 0.5 L × W^2^, where L = length and W = width. On day 35, the mice were sacrificed, and the subcutaneous tumors were harvested for histological analyses.

### 4.10. Experimental Liver Metastasis Model

A total of 3 × 10^6^ HCT116-Luc cells and 5 × 10^5^ human neutrophils stimulated with 100 nM PMA at 37 °C and 5% CO_2_ for 2 h were suspended in 100 μL sterile PBS and were injected into the splenic hilum of 7–8-week-old female KSN/slc nude mice (Japan SLC). Spleens were removed 1 min after the injection to prevent splenic tumor formation or peritoneal dissemination. For in vivo bioluminescence imaging, 1 mg of VivoGlo™ Luciferin (Promega) was injected intraperitoneally into anesthetized tumor-bearing mice 10 min before imaging. Bioluminescence from HCT116-Luc cells was monitored on day 1, 7, 14, 21, 28 and 35 post-injection, using a Xenogen IVIS system (Xenogen Corporation, Alameda, CA, USA). On day 35 post-injection, the livers were harvested for histological analyses. The animal experiments were approved by the Animal Care and Use Committee of Kyoto University.

### 4.11. Statistical Analysis

All results were confirmed using at least three independent in vitro experiments, and the data from one representative experiment are presented. All values are expressed as the mean ± standard error of the mean (SEM) or standard deviation (SD). Categorical data were determined using Pearson’s chi-squared test. The statistical significance of differences was determined with Student’s *t*-test in a cell proliferation assay, a cell migration assay, FRET-based live cell imaging and a subcutaneous tumor model. A Mann–Whitney U test was used for calculating statistical significance of differences in the experimental liver metastasis model. To determine factors associated with Cit-H3 expression and serum MPO–DNA complex expression, multivariate logistic regression analyses were used, and factors with a *p* value ≤ 0.10 were included in the model. Multivariate analyses of prognostic factors were performed using the Cox proportional hazard regression model. Survival curves were calculated according to the Kaplan–Meier method and were analyzed using the log-rank test. All analyses were two-sided, and a *p* value < 0.05 was considered statistically significant. Statistical analyses were performed using JMP Pro software, version 16.1.0 (SAS Institute Inc., Cary, NC, USA).

## Figures and Tables

**Figure 1 ijms-24-01118-f001:**
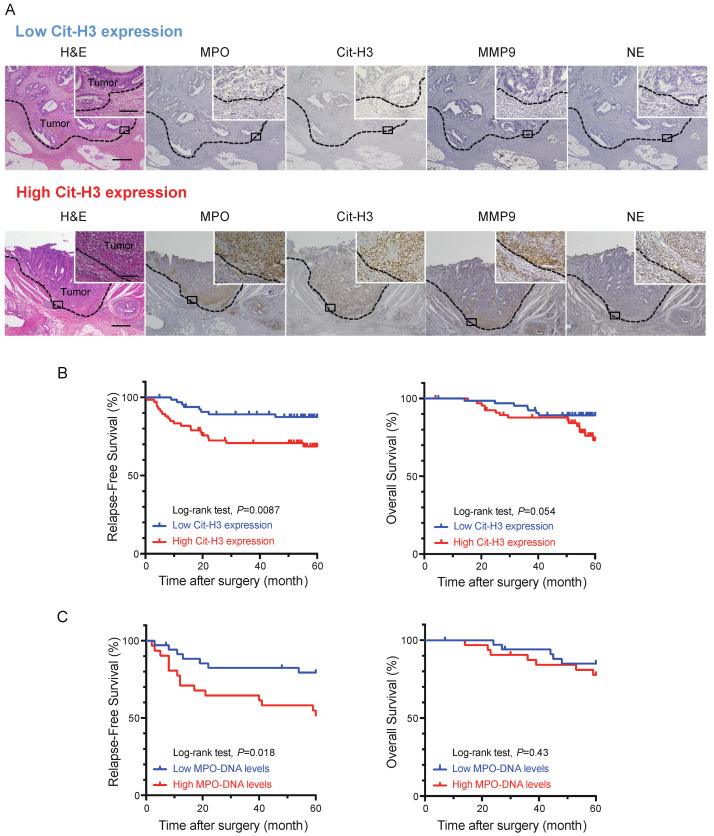
Intratumoral and systemic neutrophil extracellular trap (NET) formation correlates with poor prognoses in colorectal cancer (CRC) patients. (**A**) Hematoxylin and eosin staining (H&E) and immunohistochemical (IHC) staining for myeloperoxidase (MPO), citrullinated histon H3 (Cit-H3), matrix metalloproteinase 9 (MMP9) and neutrophil elastase (NE) in primary CRC specimens. Upper and lower panels show serial sections of representative high and low expression of Cit-H3 in CRC specimens, respectively. Scale bars, 500 μm. Magnified images are shown in right upper quadrant. Scale bars, 100 μm. (**B**) Effects of Cit-H3 expression on relapse-free survival (RFS) (right) and overall survival (OS) (left) in patients who underwent curative resection of stage I–III CRC (Kaplan–Meier estimates). *p* values were calculated using the log-rank test. (**C**) Effects of preoperative serum MPO–DNA levels on RFS (right) and OS (right) in patients who underwent curative resection of stage II and III CRC (Kaplan–Meier estimates). *p* values were calculated using the log-rank test.

**Figure 2 ijms-24-01118-f002:**
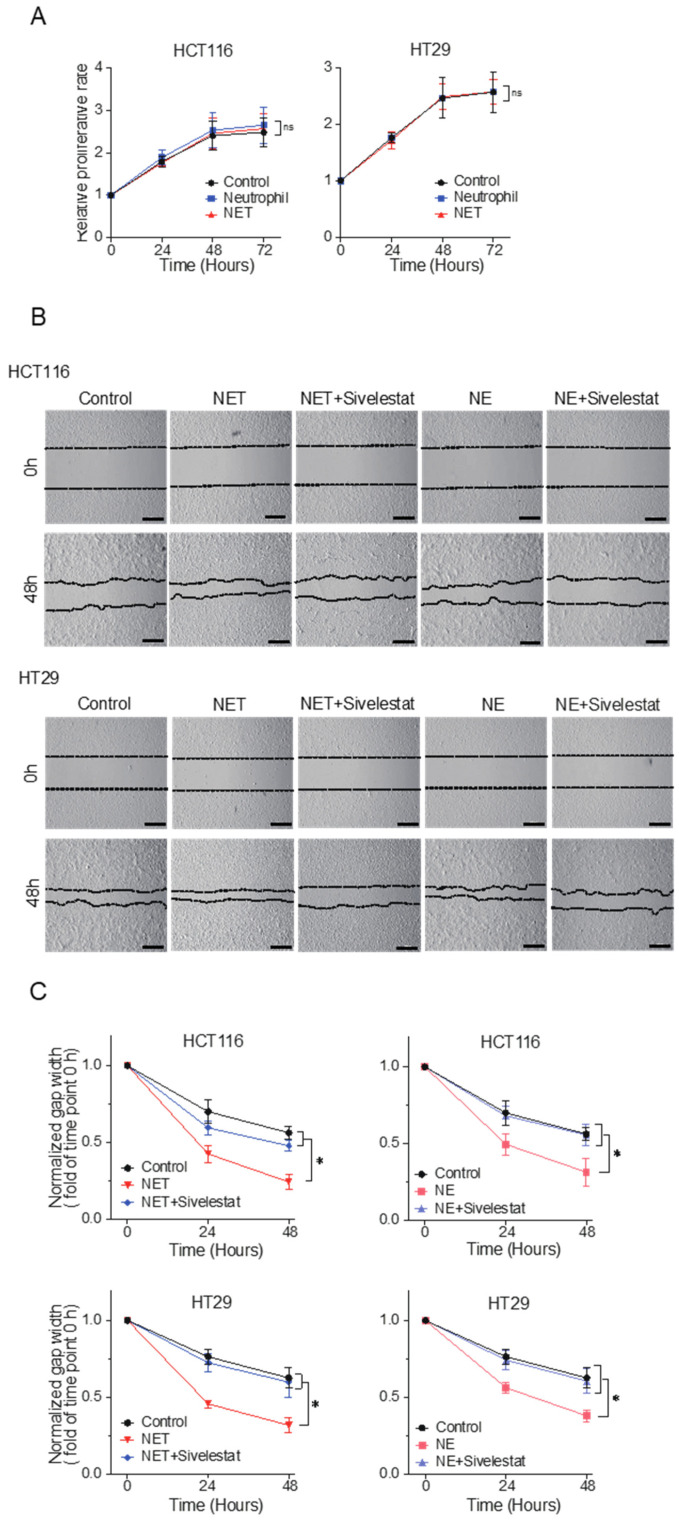
NETs accelerate the migration of CRC cells through the release of NE in vitro. (**A**) Cell proliferation assay using HCT116 and HT29 cells. Cells were incubated with NET-conditioned medium (NET-CM) or neutrophil control medium and were analyzed at 0, 24, 48 and 72 hours. The absorbance at time 0 was normalized to 1. Mean: bars ± standard error of the mean (SEM). *n* = 3. ns: not significant with Student’s *t* test. (**B**) Cell migration assay using scratch wound healing. HCT116 and HT29 cells were incubated with NET-CM or neutrophil control medium and were treated with 100 µM of sivelestat. Representative images are shown. Scale bars, 500 µm. (**C**) Wound width was measured at 0, 24 and 48 hours. Normalized gap width at hour 0 was 1. Mean: bars ± standard deviation (SD). *n* = 6. * *p* < 0.05 with Student’s *t* test.

**Figure 3 ijms-24-01118-f003:**
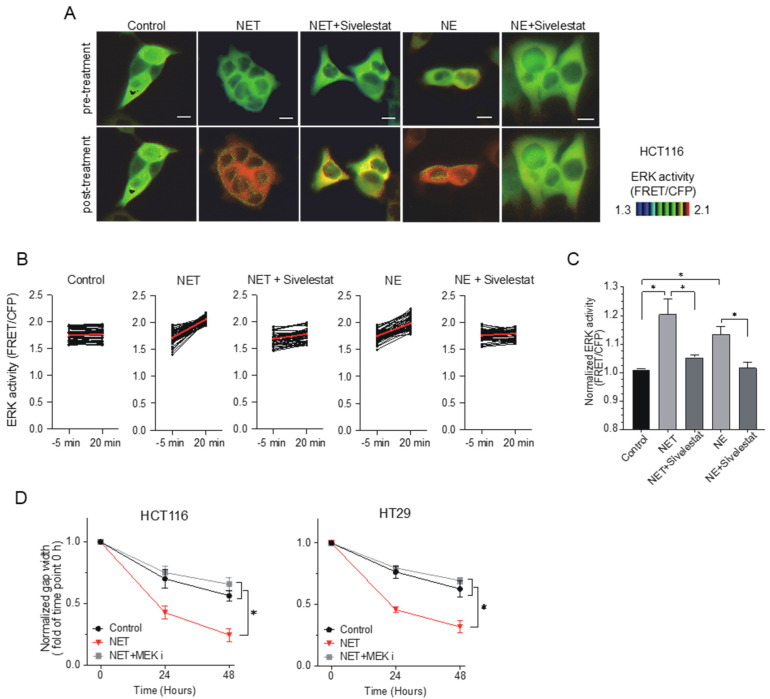
Extracellular signal-regulated kinase (ERK) is an important regulator of activated cell migration induced by NETs. (**A**). Förster resonance energy transfer (FRET)-based live cell imaging using the biosensor of ERK in HCT116 cells. HCT116 cells were serum-starved for 6 hours. Cells incubated with NET-CM were treated with 10 µg/mL of NE or 100 µM of sivelestat. Representative FRET/CFP ratio images of pre-/post-treatment are shown in the intensity-modulated display mode. Scale bars, 10 μm. (**B**). ERK activity (FRET/CFP) of pre-/post-treatment in HCT116 cells. Each dot represents the ERK activity of each cell. A total of 54 cells were analyzed. Red bars indicate the mean values. (**C**) Increased rates in normalized ERK activity (FRET/CFP) of HCT116 cells before and after the addition of NET-CM. Results are presented as the means ± SEM of triplicate measurements. * *p* < 0.05 with Student’s *t*-test. (**D**). HCT116 and HT29 cells incubated with NET-CM were treated with 10 nM of mitogen-activated protein kinase kinase (MEK) inhibitor. Wound width was measured at 0, 24 and 48 hours. The gap width at time 0 was normalized to 1. Mean: bars ± SD. *n* = 3. * *p* < 0.05 with Student’s *t* test.

**Figure 4 ijms-24-01118-f004:**
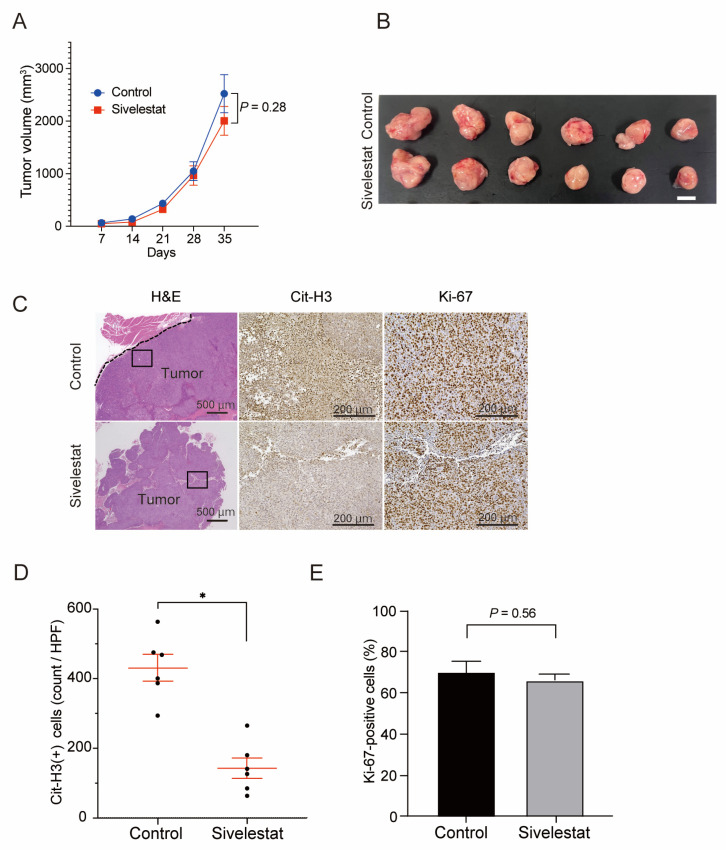
The suppression of NET formation by NE inhibitors had no effect on the growth of subcutaneous implanted tumors of CRC cells in vivo. (**A**) Growth curves of the subcutaneous transplanted tumors of HCT116 in the KSN/slc nude mice treated with sivelestat (10 mg/kg) or vehicle control. Mean: bars ± SEM. *n* = 6 tumors in each group. *p* values calculated with Student’s *t*-test. (**B**) Representative macroscopic images of the subcutaneous transplanted tumors of HCT116 cells dissected from KSN/slc nude mice treated with sivelestat (10 mg/kg) or vehicle control on day 35. Scale bars, 10 mm. (**C**) IHC staining for H&E, Cit-H3 and Ki-67 in subcutaneous transplanted tumors. Representative images are shown (H&E: scale bars, 500 μm. Cit-H3, Ki-67: scale bars, 200 μm). (**D**,**E**). Quantification of Cit-H3-positive cells (**D**) and percentage of Ki-67-positive cells (**E**) in the IHC staining of xenograft tumors. Mean: bars ± SEM. *n* = 6 tumors in each group. * *p* < 0.05 with Student’s *t*-test.

**Figure 5 ijms-24-01118-f005:**
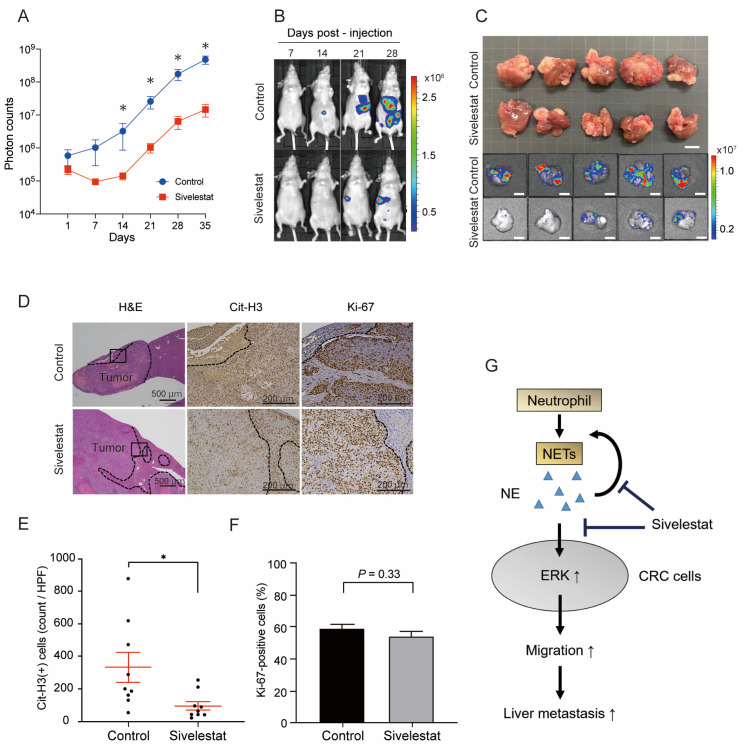
The suppression of NET formation by NE inhibitors decreased liver metastases. (**A**) Quantification of HCT116-Luc liver metastatic lesions (photon counts) in KSN/slc nude mice treated with sivelestat (10 mg/kg) or vehicle control. Mean: bars ± SEM. *n* = 9 mice in each group. * *p* < 0.05 with Mann–Whitney U test. (**B**). Representative in vivo bioluminescence images of HCT116-Luc liver metastases in KSN/slc nude mice treated with sivelestat (10 mg/kg) or vehicle control. (**C**) Representative macroscopic and bioluminescence images of the HCT116-Luc liver metastatic tumors dissected from KSN/slc nude mice treated with sivelestat (10 mg/kg) or vehicle control on day 35. Scale bars, 10 mm. (**D**). IHC staining for H&E, Cit-H3 and Ki-67 in experimental liver metastasis. Representative images are shown. Scale bars, 500 μm for H&E, 200 μm for Cit-H3 and Ki-67 staining. E and F. Quantification of Cit-H3-positive cells (**E**) and percentage of Ki-67-positive cells (**F**) in the IHC staining of xenograft tumors. Mean: bars ± SEM. *n* = 9 tumors in each group. * *p* < 0.05 with Student’s *t*-test. (**G**) Schematic representation of the possible mechanism by which NETs promote the liver metastasis of CRC cells. NE released during NETosis increases ERK activity, which accelerates the migration of CRC cells. Inhibition of NE suppresses the NETosis and the migration of CRC cells resulting in decreased liver metastases.

**Table 1 ijms-24-01118-t001:** Univariate and multivariate analysis using clinicopathological characteristics and Cit-H3 expression for RFS in 133 CRC patients.

Variables	Univariate Analysis			Multivariate Analysis		
	HR	95% CI	*p*	HR	95% CI	*p*
Age, years (<68 vs. ≥68)	1.02	0.48–2.17	0.96			
Sex (male vs. female)	0.57	0.24–1.35	0.2			
Location (colon vs. rectum)	1.99	0.95–4.19	0.069	1.71	0.75–3.92	0.2
T factor (Tis-T2 vs. T3-T4)	1.99	0.76–5.24	0.16			
* n * factor (negative vs. positive)	4.86	2.2–10.77	< 0.0001	3.26	1.38–7.70	0.0071
Lymphatic invasion (ly0 vs. ≥ly1)	3.69	1.67–8.17	0.0013	2.01	0.84–4.80	0.12
Venous invasion (v0 vs. ≥v1)	2.03	0.92–4.48	0.08	1.33	0.57–3.08	0.51
Preoperative therapy (no vs. yes)	4.68	1.77–12.37	0.0019	2.04	0.67–6.21	0.21
Adjuvant therapy (no vs. yes)	1.35	0.64–2.84	0.43			
NLR (<3 vs. ≥3)	1.05	0.47–2.32	0.91			
CEA (ng/mL, <5 vs. ≥5)	1.34	0.63–2.81	0.44			
CA19-9 (U/mL, <37 vs. ≥37)	1.94	0.73–5.11	0.21			
Cit-H3 expression (low vs. high)	2.86	1.26–6.49	0.012	2.86	1.25–6.54	0.013

NLR, neutrophil lymphocyte ratio; CEA, carcinoembrionic antigen; CA19-9, carbohydrate antigen 19-9; HR, hazard ratio; CI, confidence interval.

**Table 2 ijms-24-01118-t002:** Univariate and multivariate analysis using clinicopathological characteristics and serum MPO–DNA levels for RFS in 67 CRC patients.

Variables	Univariate Analysis			Multivariate Analysis		
	HR	95% CI	*p*	HR	95% CI	*p*
Age, years (<68 vs. ≥ 68)	0.58	0.23–1.42	0.23			
Sex (male vs. female)	0.46	0.18–1.14	0.094	0.26	0.10–0.72	0.009
Location (colon vs. rectum)	0.51	0.18–1.39	0.19			
T factor (Tis-T3 vs. T4)	1.32	0.44–3.96	0.62			
* n * factor (negative vs. positive)	1.64	0.68–3.95	0.27			
Lymphatic invasion (ly0 vs. ≥ ly1)	2.1	0.76–5.80	0.18			
Venous invasion (v0 vs. ≥ v1)	3.66	1.07–12.5	0.038	5.69	1.58–20.51	0.008
Preoperative therapy (no vs. yes)	1.19	0.28–5.14	0.81			
Adjuvant therapy (no vs. yes)	0.94	0.39–2.26	0.89			
NLR (<3 vs. ≥ 3)	1.6	0.66–3.87	0.29			
CEA (ng/mL, <5 vs. ≥5)	1.33	0.55–3.21	0.53			
CA19-9 (U/mL, <37 vs. ≥ 37)	4.08	1.46–11.43	0.007	3.16	1.07–9.22	0.036
MPO–DNA levels (low vs. high)	2.35	0.94–5.90	0.06	3.53	1.31–9.56	0.013

MPO, Myeloperoxidase.

## Data Availability

The data that support the findings of this study are available on request from the corresponding author.

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
