# Peer review of "Neutrophil Extracellular Traps Promote Metastases of Colorectal Cancers through Activation of ERK Signaling by Releasing Neutrophil Elastase"

_ijms, 2023, doi:10.3390/ijms24021118_

Round 1
Reviewer 1 Report
The manuscript by Okamoto et al., submitted for publication is an in vivo and in vitro study investigating the clinical relevance and molecular mechanism of NETs in CRCs. This study confirmed that high expression of NETs correlates with poor RFS and inhibition of neutrophil elastase in a liver metastatic mouse model decreased of liver metastases of CRC cells. The study is well designed and the manuscript is well organized and well written and fairly easy for the reader to follow. Although there are several typos and the narrative is a kind of fragmentary, I just abide by scientific soundness. I would like to offer the following points for consideration by the authors towards the improvement of the manuscript:
1- In the last paragraph of the introduction, please state the aims of the study rather than the findings of study.
2- Because you included patients with stage 3 CRC in the study, please include whether or not they received adjuvant therapy for univariate and multivariate analysis for RFS and OS.
3- Please specify how you divide the patients into two(high-low) according to Cit-H3 expression? (median value or ROC analysis ?)
4- How was the number of animals determined? Was there a power calculation?
5- How were the dose of sivelestat determined?
6- Please specify the limitations of the study in more detail and in the paragraph before the conclusion.
Author Response
Reviewer1
1- In the last paragraph of the introduction, please state the aims of the study rather than the findings of study.
>> We appreciate for the great suggestion. We have added the aims of our study (Line 85-87) and modified the description in the introduction (Line 90-91).
2- Because you included patients with stage 3 CRC in the study, please include whether or not they received adjuvant therapy for univariate and multivariate analysis for RFS and OS.
>> We agree with this comment. We re-analyzed the cohort and added the data comparing the groups with or without adjuvant chemotherapy (table 1, 2, S1, S2, S3, and S4).
3- Please specify how you divide the patients into two(high-low) according to Cit-H3 expression? (median value or ROC analysis ?)
>> We divided the patients into two groups according to the median value. We have added to the description in materials & methods (Line 441).
4- How was the number of animals determined? Was there a power calculation?
>> We did not calculate the statistical power to determine the number of experimental animals. Based on the following points, we determined the sample size in this study.
(1) The outcome is a continuous variable.
(2) This study is an exploratory study and it is difficult to predict the results in advance.
(3) From the viewpoint of animal welfare, it is recommended to perform experiments with a minimum number of experimental animals by the Animal Care and Use Committee of Kyoto University.
5- How were the dose of sivelestat determined?
>> The dose of sivelestat was determined according to the previous studies.
1: DOI:10.1111/j.1349-7006.2006.00278.x. (Reference #74)
2: DOI:10.1530/ERC-19-0431. (Reference #75)
3: DOI:10.1016/j.biomaterials.2020.119836. (Reference #76)
4: DOI:10.1111/apm.12222. (Reference #80)
We have added the description in materials and methods section (Line 498-499 and line 538-539) and cited references (Reference #74, #75, #76, and #80). We apologize for the misstatement regarding the unit of sivelestat concentrations. We have corrected the unit from nM to µM in the manuscript (Line 195, 219, 496, 498, and 513).
6-  Please specify the limitations of the study in more detail and in the paragraph before the conclusion.
> We have added the following description about the limitations of our study in the paragraph before the conclusion. (Line 403-407)
“This study has several limitations. First, in clinical analysis, patients were recruited from a single center and the sample size was relatively small. Second, we focused on the effect of NE among the granular proteins released during the process of NET formation. Additional analyses of the effects of other granular proteins on CRC would be necessary to reveal the detailed mechanisms by which NETs promote CRC progression.” (Line 403-407)

Reviewer 2 Report
Title: “Neutrophil extracellular traps promote metastases of colorectal cancersthrough activation of ERK signaling by neutrophil elastase release” Authors: Michio Okamoto, Rei Mizuno, Kenji Kawada, Yoshiro Itatani,
Yoshiyuki Kiyasu, Keita Hanada, Wataru Hirata, Yasuyo Nishikawa, Hideyuki
Masui, Naoko Sugimoto, Takuya Tamura, Susumu Inamoto, Yoshiharu Sakai,
Kazutaka Obama
Summary:
This article presents both in vitro (HCT116, HT29) and in vivo (mouse model) experiments suggesting that NETs activate the ERK signaling pathway and thus contribute to colorectal cancer metastasis. Extensive experiments were performed and presented.
I propose the following additions:
1) Abstract, line 30/31: I would mention the cell types used as well as the mouse species in parentheses respectively.
2) Introduction:
(a) line 52: Please add why the TME is tumor-promoting (cross-talk, inflammation, NF-kB, TNF, CD1, FAK, MMP, CAFs...).
Please add an additional reference:
doi: 10.1002/path.6011,
doi: 10.3390/molecules25184292,
doi: 10.1186/s12967-022-03510-8,
doi: 10.3389/fonc.2021.650603,
doi: 10.3390/ijms23094714.
(b) Line 68/74: NE and ERK are only enumerated here. Since these terms appear in the title and are thus main features of the article, they should be briefly explained.
3) Results:
(a) Please remove the dots at the end of headings 2.1.-2.4.
b) If parameters are mentioned, it would be helpful for the reader to mention the meaning of the value, for example that CA19-9 and CEA are tumor markers.
c) Chapter 2.4/Figure 4B: Please add how long was the total duration of the vivo experiment and on which day of the experiment the collected samples were taken.
4) Discussion/Conclusions: A summary diagram of the results would be helpful for the reader to get an overview.
5) Materials and Methods:
(a) 4.1.: Please add the distribution of age and sex of the patients.
(b) 4.7.: Please add which technique and instrument was used to measure wound width and which site was assessed (narrowest/widest site or an average?).
c) 4.11.: Highlight here the evaluation of the in vitro experiments. How was the evaluation of the in vivo experiments performed?
Author Response
Reviewer2
1) Abstract, line 30/31: I would mention the cell types used as well as the mouse species in parentheses respectively.
>> The cell types and mouse species were added to abstract with parentheses (Line 30 and 32-33).
2) Introduction:
(a) line 52: Please add why the TME is tumor-promoting (cross-talk, inflammation, NF-kB, TNF, CD1, FAK, MMP, CAFs...).
>> As requested, we added the following description citing the references indicated by the reviewer (Line 53 - 55).
“The cross-talk between cancer cells and the components of TME mediated by TGF-b, TNF-a, TNF- b, and NF-kB signaling contribute to the cancer progression.” (Line 53 - 55)
(b) Line 68/74: NE and ERK are only enumerated here. Since these terms appear in the title and are thus main features of the article, they should be briefly explained.
>> Brief explanations of NE (line72-74) and ERK (line 79-82) has been added to the introduction.
3) Results:
(a) Please remove the dots at the end of headings 2.1.-2.4.
>> As requested, dots were removed (Line 96, 163, 199 and 230).
(b) If parameters are mentioned, it would be helpful for the reader to mention the meaning of the value, for example that CA19-9 and CEA are tumor markers.
>> As requested, we have added the explanation about the meaning of parameter (Line 132).
(c) Chapter 2.4/Figure 4B: Please add how long was the total duration of the vivo experiment and on which day of the experiment the collected samples were taken.
>> The duration of the treatment for the subcutaneous tumor model was added in chapter 2.4, Figure 4B and materials & methods (Line 241, 254, 538, and 542-543).
4) Discussion/Conclusions: A summary diagram of the results would be helpful for the reader to get an overview.
>> We appreciate for the great suggestion. We have added the schematic representation in Fig.5 and explanation of schema was added to the figure legend. (Line 281 and 294-297)
5) Materials and Methods:
(a) 4.1.: Please add the distribution of age and sex of the patients.
>> The distributions of age and sex of the patients were stated in table S1 and S3.
(b) 4.7.: Please add which technique and instrument was used to measure wound width and which site was assessed (narrowest/widest site or an average?).
>> The wound was observed with a phase-contrast microscope (BZ-X810; Keyence) to measure the widths of six points in each wound. The averaged value of these six points was used as a width of each wound. We have added the description in the materials and methods (Line 499-501).
(c) 4.11.: Highlight here the evaluation of the in vitro experiments. How was the evaluation of the in vivo experiments performed?
>> The statistical significance of differences was determined by Student’s t-test in cell proliferation assay, Cell migration assay, FRET based live cell imaging and subcutaneous tumor model. Mann-Whitney U test was used for calculating statistical significance of differences in experimental liver metastasis model. (Line 561-564)

Round 2
Reviewer 1 Report
I am satisfied that the authors have addressed all of my previous concerns about the article. It is now much improved and I feel that it is now suitable for publication.